# Object Coreference in German: The Reflexive *sich* as a Problem for Derivational Approaches to Binding

**Vera Lee-Schoenfeld** [1],* and **Nicholas Twiner** [2]

1    Department of Linguistics, Franklin College of Arts and Sciences, University of Georgia, Athens, GA 30602, USA

2    Department of Linguistics, Stanford University, Stanford, CA 94305, USA; nftwiner@gmail.com

*    Correspondence: vleesch@uga.edu

**Abstract:** Despite Grewendorf's well-known German binding data with the double-object verb *zeigen* 'show', where one object reflexively binds the other and which suggests that the direct object (DO) is generated higher than the indirect object (IO), this paper argues for the canonical surface order of IO > DO as base order. We highlight the exceptional status of Grewendorf's examples, build on scope facts as well as a quantitative acceptability rating study, and exploit the fact that *zeigen* can also be used as inherently reflexive with idiomatic meaning. Appealing to the base configuration of the pieces of idiomatic expressions and considering different Spell-Out possibilities of coreferential objects in German, we show that the case, number, and gender underspecification of the anaphor *sich* poses a previously unnoticed problem for derivational approaches to binding.

**Keywords:** reflexive binding; double-object construction; object coreference; structural accusative case; inherent dative case; inherent reflexivity; subject orientation

## 1. Introduction

The base order of internal arguments in German double object constructions has been argued to be determinable by binding facts (see, e.g., [1–3]). For instance, given Grewendorf's famous examples in (1) [4] and assuming that the order of internal arguments satisfies standard binding conditions [5,6], it has been argued, in line with grammatical function hierarchies like SUBJ > DO > IO > ... (see, e.g., [7]), that the accusative (ACC) direct object (DO) must be generated above the dative (DAT) indirect object (IO).

(1)  a.  Der    Arzt    zeigte   den       Patienten$_i$  sich$_i$/*ihm$_i$          im      Spiegel.
         the    doctor  showed   the.ACC   patient        himself.DAT/him.DAT       in.the  mirror
     b.  Der    Arzt    zeigte   dem       Patienten$_i$  *sich$_i$/ihn$_i$          im      Spiegel.
         the    doctor  showed   the.DAT   patient        himself.ACC/him.ACC       in.the  mirror
         'The doctor showed the patient himself in the mirror.'

The canonical surface order of German internal arguments (IO > DO) in other examples would then have to be derived via obligatory scrambling of the IO above the DO (into a DAT-case-licensing A'-position, according to [1]). A main goal of this paper is to argue against conclusions along these lines.

We start by revisiting Grewendorf's data in Section 2. Then, in Section 3, we provide support for the canonical German surface order of IO > DO as base order. Section 4 explains how interference of inherently reflexive readings[1] and the nature of the exceptional mirror image scenario in (1) can lead to DO > IO order. Finally, in Section 5, we consider a derivational approach to binding allowing for different Spell-Out possibilities and thereby avoiding interference from the inherently reflexive use of ditransitive verbs. The ability of *sich* to find more than one binder, however, turns out to be at odds with derivational approaches to binding. In Section 6, we conclude that, in part due to its underspecification for case, number, and gender, *sich* has different binding possibilities – it can be, and usually

is, subject-oriented, but it can also be object-oriented. Interference of one binding possibility with the other is unexpected on the view that a reflexive pronoun is (a part of) one and the same nominal as its antecedent underlyingly. This clearly speaks against a derivational binding account (as proposed, e.g., in [9–12]) of German object coreference.

## 2. Shedding Light on Grewendorf's Mirror Image Data

For native speakers, who have not read about examples like (1a–b) in the literature, the first and only possible reading of (1a) that comes to mind is that the doctor showed himself to the patients (plural!) in the mirror, as in (1′).

(1') Der Arzt$_i$ zeigte den Patienten sich$_i$ im Spiegel.
the doctor$_i$ showed the.**DAT** patient**s** himself$_i$ in.the mirror
'The doctor showed the patients himself in the mirror.'

Here, the anaphor *sich*, which is uninflected for case, number, and gender, is referring to the subject, as expected, given that reflexive pronouns typically are subject-oriented. The non-anaphoric object *den Patienten* is understood not as ACC singular masculine but as DAT plural. To eliminate the syncretism involved with these two forms and thereby force speakers to interpret the anaphor as getting its reference from the object (and also to present the verb and its arguments in their base order), we changed Grewendorf's examples as shown in (2), where *Patientin* is singular feminine (F).

(2) a. dass der Arzt$_i$ die Patientin$_j$ sich*$_i$/$_j$/ihr*$_j$ im Spiegel zeigte.
that the.NOM doctor$_i$ the.ACC patient.F$_j$ REFL*$_i$/$_j$/her*$_j$.DAT in.the mirror showed
'that the male doctor showed the female patient herself in the mirror.'

b. dass der Arzt$_i$ der Patientin$_j$ sich$_i$/*$_j$/sie$_j$ im Spiegel zeigte.[2]
that the.NOM doctor$_i$ the.DAT patient.F$_j$ REFL$_i$/*$_j$/her$_j$.ACC in.the mirror showed
'that the male doctor showed {himself to the female patient / the female patient$_j$ her$_j$ in the mirror}.'

The non-anaphoric object, *die Patientin* in the (a)-example is now unambiguously ACC-marked, which has the welcome consequence that speakers interpret the anaphor *sich* as being able to refer to only the object in the (a)-example and only the subject in the (b)-example. Still, speakers tend to want to rephrase (2a) entirely in order to express the intended meaning. This confirms that object orientation of the anaphor is a very marginal possibility that speakers generally avoid.

What adds to the marginality of Grewendorf's data is the order of DO(ACC) > IO(DAT) because the unmarked order of objects in constructions involving a ditransitive verb is the opposite, IO(DAT) > DO(ACC), as in *jemandem etwas geben* 'give somebody.DAT something.ACC'. Also, taking a step back from the morpho-syntax of these sentences, it is worth noting that the situation of showing people themselves in the mirror is rather unusual. People are either shown something or someone other than themselves or, if they look into a mirror, no third party is involved. The one setting where this mirror image scenario might be considered normal is a hair salon: The hair stylist looks at and talks to the client in the mirror and shows them their hair, so that the mirror image, is treated like the actual person, the recipient, and the actual person whose hair is being shown, is treated like the mirror image, the theme—an interesting role reversal that we return to in Section 4. First, in Section 3, we examine double object binding data involving ditransitive verbs other than *zeigen* 'show' and reconstruction effects to show that there is ample evidence for IO > DO, as opposed to DO > IO, as base order.

## 3. Evidence against DO > IO and for IO > DO as Base Order

Notice that constructions with classic ditransitive verbs like *schicken* 'send' and *schenken* 'give as a gift', as well as *empfehlen* 'recommend', which lends itself more naturally to object coreference involving animate entities, do not pattern like Grewendorf's.

(3)  a.  *dass    wir  die   Sängerin<sub>i</sub>  sich<sub>i</sub>      als  Wachsfigur  schicken  wollten.
         that     we   the.ACC singer.FEM  REFL.DAT  as  wax.figure  send      wanted
         intended: 'that we wanted to send the singer herself as a wax figure.'
     b.  *dass    ich  meinen Vater<sub>i</sub>  zum  Geburtstag  sich<sub>i</sub>      als Statue  geschenkt  habe.
         that     I    my.ACC father  for.the birthday   REFL.DAT as statue  given      have
         intended: 'that I gave my dad himself as a statue for this birthday.'
     c.  *dass    man  die    Angeklagte<sub>i</sub>  sich<sub>i</sub>      als Anwältin  empfohlen     hat.
         that     one  the.ACC accused     REFL.DAT as attorney    recommended has
         intended: 'that people recommended to the accused herself as the attorney.'

In all of (3a–c), DO(ACC) > IO(DAT) is ungrammatical, and, importantly, all these examples get better if ACC and DAT case marking on the objects is switched, so that the order is IO(DAT) > DO(ACC), especially when *sich* is intensified with *selbst* 'self'.

Furthermore, scope reconstruction effects strongly suggest that the base order of arguments in non-reflexive contexts is IO > DO, not DO > IO (see, e.g., [13,14]). Assuming that a quantifier can be interpreted either in its surface or its base position, we expect it to cause scope ambiguity if it moves from a position lower than another quantifier to a position higher than this other quantifier. Likewise, if the moving quantifier originates higher than the other quantifier, we do not expect scope ambiguity. These expectations are borne out in (4) and (5), respectively.

(4)  Genau  **einen**  Gast   hat  sie  **jedem**    Freund  vorgestellt  (*einen > jedem; jedem > einen*)
     exactly  **one.ACC** guest  has  she  **each.DAT** friend  introduced
     [15] (p. 242)

(5)  Genau  **einem**  Freund  hat  sie  **jeden**   Gast   vorgestellt  (*einem > jeden*)
     exactly  **one.DAT** friend  has  she  **each.ACC** guest  introduced
     [15] (p. 241)

Example (4) is ambiguous, the interpretations being that (i) there was one guest who was introduced to every friend or (ii) for every friend, there was a potentially different guest who was introduced to this friend. Example (5), on the other hand, is unambiguous, the only possible interpretation being that there was one friend to whom every guest was introduced. Thus, in (4), where the ACC-marked quantificational DP has been topicalized, it takes scope over the DAT-marked quantifier only in its landing site, not in its origin site, while, in (5), where the DAT-marked quantificational DP has been topicalized, it takes scope over the lower ACC-marked quantifier in both its origin and its landing site. This leads us to conclude that, in their base positions, the DAT-marked IO must be structurally higher than, i.e., must c-command, the ACC-marked DO, yielding IO > DO as base order.

Finally, a quantitative study by Featherston and Sternefeld, an acceptability rating experiment [16], suggests that Grewendorf's (1a)/our (2a) is only one of several possible double object formulations German speakers use to express 'showing someone to themselves' and that it is a rather exceptional one. The study produces three relevant generalizations about the Spell-Out possibilities for object coreference. They are given in (6a–c).

(6)  a.  DAT antecedents are more accepted than ACC antecedents.
     b.  Reflexives are more accepted than non-reflexive pronouns as anaphoric
         elements.
     c.  Speakers prefer use of the intensifier *selbst* with both reflexive and non-
         reflexive pronouns.

Although these generalizations are just tendencies that may exist for a number of reasons not taken into account here, we would still like to point out that, if greater grammaticality is taken as an indication of underlying syntactic structures, generalization (a) suggests the following: Antecedents originate in (rather than move to) the inherent DAT-licensing position (contra [1]), and, since DAT marks IOs, the antecedent should be the IO, and the anaphoric element, the DO. Generalizations (a) and (b) combined suggest that the DO is a

reflexive rather than a non-reflexive pronoun, and since reflexives must be c-commanded by their antecedents, we arrive at the order of IO > DO (see [3] for a different take on [16]'s findings).[3] Consistent with our data in (3), the scope reconstruction effects in (4–5), and generalizations (6a–b), we argue for the structure in Figure 1 as the base configuration of the verbal argument domain. This structure has also been independently motivated by [17–19].

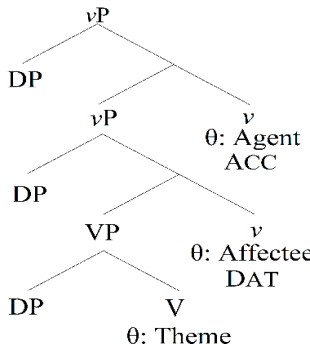

**Figure 1.** Base structure of the verbal domain including Agent, Affectee[4] (IO), and Theme (DO).

Each verbal head assigns its theta-role to the DP in its projection. Furthermore, the case feature listed under each of the *v*-heads values the case of a certain DP as follows: Affectee *v* licenses inherent DAT on the Goal/Recipient DP (IO), agentive *v* licenses structural ACC on the Theme/Patient DP (DO), and T licenses structural NOM on the Agent DP (SUBJ).

### 4. The Exceptional Status of Object Coreference in (1a)/(2a): Interference from Inherent Reflexivity

The question is why the ACC-marked antecedent is grammatical in Grewendorf's (1a)/our (2a), with the verb *zeigen* 'show', but not in the other object coreference examples in (3), with the verbs *schicken* 'send', *schenken* 'give (as a gift)', and *empfehlen* 'recommend'.

Notice that *zeigen* can be used to express two different meanings: (i) 'show someone something', which corresponds to the ditransitive use of the verb, *jemandem etwas zeigen*, and (ii) 'let oneself be seen (by someone)/appear (in public)', which corresponds to the inherently reflexive, idiomatic use of the verb, *sich (jemandem) zeigen*, with an optional DAT argument. Meaning (ii) is shown in (7), where *sich* can occur higher (in parentheses) or lower (not in parentheses).

(7)  dass    (sich)      die      Königin    sich       der     Menge     zeigte.
     that    (REFL)      the.NOM  queen      REFL.ACC   the.DAT crowd     showed
     'that the queen appeared to the crowd.'

Meaning (ii) is the only readily available meaning expressed by Grewendorf's (1b)/our (2b): the anaphor is ungrammatical when referring to the DAT-marked object but grammatical when referring to the subject. The higher position of *sich* in (7) as an alternative to the lower one is also an option in (2b), repeated here as (2'b).

(2')  b.  dass    (sich)   der      Arzt     der      Patientin  sich       im       Spiegel   zeigte.
         that    (REFL)   the.NOM  doctor   the.DAT  patient.F  REFL.ACC   in.the   mirror    showed
         'that the male doctor showed himself to the female patient in the mirror.'

This suggests that we are dealing with the inherently reflexive use of *zeigen* in these examples. Importantly, in its pre-subject position, *sich* cannot bear stress or be intensified with *selbst*, as shown in (2''b).

(2'')  b.  *dass   sich    selbst   der      Arzt     der      Patientin  im       Spiegel   zeigte.
          that    REFL    self     the.NOM  doctor   the.DAT  patient.F  in.the   mirror    showed
          'that the male doctor showed himself to the female patient in the mirror.'

Crucially, the verb *zeigen* in Grewendorf's examples is most naturally interpreted as inherently reflexive, with a DAT-marked non-reflexive object and subject-orientation of the anaphor *sich*, and this holds regardless of ACC > DAT or DAT > ACC order, that is, whether *sich* shows up higher or lower than the other object.

Based on everything laid out thus far, our hypothesis is as stated in (8).

(8) **Hypothesis**: The order of DO(ACC) > IO(DAT) in (1a)/(2a) is only acceptable because the preferred order of IO(DAT) > DO(ACC) in double-object constructions resembles the inherently reflexive use (meaning ii) of the verb in that the non-reflexive object is DAT-marked, and when this meaning is not intended, DO(ACC) > IO(DAT) appears to be the best alternative, where the non-reflexive object is ACC-marked.

Interestingly, given the hair salon mirror image scenario (see Section 2), this alternative even works semantically. The mirror image (normally the ACC-marked DO, i.e., what's shown, the Theme) is treated like the actual person, and the actual person (normally the DAT-marked IO, i.e., the Recipient of the showing) is treated like the mirror image, so that the roles of Goal/Recipient and Patient/Theme are reversed, leading to DAT-ACC case reversal as well. Vogel [3] (p. 376) would argue against this because his claim is that, when antecedent and anaphor refer to different entities (like the actual person and the wax figure of this person), "only the real person may be the antecedent and the statue/image the bound element, not the other way around". He calls this the "Ringo constraint". To support his claim and equating wax figures with mirror images, he provides the examples in (9) (taken from [20]) and (10).

(9)   a.   All of a sudden Ringo started undressing himself.
         (*himself* = person or statue)

      b.   All of a sudden I accidentally bumped into the statues, and
         *Ringo toppled over and fell on himself.
         (*Ringo* = statue; *himself* = person)
         [20] (p. 4)

(10)  a.   I showed John himself in the mirror.
      b.   *I showed John to himself in the mirror.
         [3] (p. 376)

(10b) is supposed to be a Ringo constraint violation because the antecedent (*John*) is the mirror image, and the anaphor (*himself*) is the real person. Contra Vogel, we argue that (10b) is perfectly fine given the hair salon scenario.[5] A mirror image is much more like the actual person than a wax figure and therefore escapes the Ringo constraint.

*Ditransitive Verbs besides Zeigen*[6]

If there are other exceptional ditransitive verbs like *zeigen* 'show', which allow object coreference with DO(ACC) > IO(DAT) order but without involving a mirror image situation, the hair-salon-induced thematic role and case reversal cannot be the whole story. This brings us back to our hypothesis in (8), i.e., interference from inherent reflexivity.

In this subsection, we walk the reader through examples with several other ditransitive verbs that allow for object coreference with DO(ACC) > IO(DAT) order. We conclude that what they all have in common is an inherently reflexive use and that this is what leads to the non-canonical order of DO(ACC) > IO(DAT).

The other ditransitive verb (besides *zeigen* 'show') that shows up in Grewendorf's examples [4] which suggest that object coreference is only possible given DO(ACC) > IO(DAT) order is ***vorstellen*** 'introduce'. Its (di)transitive use, which we will label (i), (*jemandem*.DAT) *jemanden/etwas*.ACC *vorstellen* 'introduce someone/something (to somebody/one another)', has the canonical IO > DO order as its unmarked order. It is really a transitive verb with an optional DAT argument. This is illustrated in (11).

(11)　dass der　　　Junge bei der Feier　　(seinen Eltern)　seine　Freundin vorstellte.
　　　that the.NOM boy　at the party　　(his.DAT parents)　his.ACC girl.friend introduced
　　　'that the boy introduced his girlfriend (to his parents) at the party.'

Its other uses are inherently reflexive: (ii) *sich (jemandem/einander*.DAT*) vorstellen* 'introduce oneself/say one's name (to someone/one another)', with an optional DAT argument, as in (12):

(12)　dass sich die　　　Professorin　(den　　　Studierenden)　vorstellt.
　　　that REFL the.NOM professor-FEM　(the.DAT students)　　introduces
　　　'that the professor is introducing herself (to the students).'

and (iii) *sich etwas*.ACC *vorstellen* 'imagine something', as in (13):

(13)　dass sich der　　Junge　so etwas　　　nicht vorstellen kann.
　　　that REFL the.NOM boy　such a.thing.ACC　not　imagine　can
　　　'that the boy cannot imagine something like that'

Notice the pre-subject position of *sich* in both (12) and (13), supporting the analysis of uses (ii) and (iii) of the verb as being inherently reflexive. In (11), the order of the DAT and ACC-objects can, of course, be switched, but this does not speak against IO > DO as base order and is to be expected in a language where scrambling motivated by information structure is quite common (see, e.g., [21,22]).

However, Grewendorf's examples in (14) [4] and Vogel's examples in (15) [3], where only DO > IO is grammatical when a reciprocal or reflexive is involved, are problematic in that they seem to fall into the category of Grewendorf's (1a)/our (2a).

(14)　a.　dass　　man　　　die　　Gäste_i　einander_i　　　　vorgestellt hat.
　　　　　　that　　one.NOM　the.ACC guests_i　one-another.DAT_i　introduced has
　　　　　　'that the guests were introduced to each other.'
　　　b.　*dass　man　　　den　　Gästen_i　einander_i　　　　vorgestellt hat.
　　　　　　 that　one.NOM　the.DAT guests_i　one-another.ACC_i　introduced has
　　　　　　'that the guests were introduced to each other.'

(15)　a.　Ich habe die　　Gäste_i　sich_i　　　gegenseitig vorgestellt.
　　　　　　I　have the.ACC guests_i　REFL.DAT_i each-other　introduced
　　　　　　'I introduced the guests to each other.'
　　　b.　*Ich habe den　　Gästen_i　sich_i　　　gegenseitig vorgestellt.
　　　　　　 I　have the.DAT guests_i　REFL.ACC_i each-other　introduced
　　　　　　'I introduced the guests to each other.'

Assuming that the reciprocal *einander* functions as an anaphor, Grewendorf treats it just like the reflexive *sich* and therefore takes examples (14a–b) to make the same point as his examples (1a–b)/our (2a–b), namely that only the order of DO(ACC) > IO(DAT) is possible when anaphoric binding is involved.

We side with Sternefeld and Featherston [15], who show that *einander* is in fact not an anaphoric argument but just an adjunct that can be added to the (di)transitive uses of *vorstellen* (i) and (ii). In … *dass sich_i die Gastgeber_i mir einander vorstellten* 'that the hosts introduced one another to me', for instance, the reflexive *sich*, the DAT *mir*, and the reciprocal *einander* all co-occur. Given the high position of *sich*, we know that we are dealing with the inherently reflexive use of the verb. Besides the ACC *sich*, the DAT *mir* must then be the one optional argument this use of the verb allows, and *einander* can only be an adjunct added to clarify that the hosts did not introduce themselves but one another (A introduced B to me, and B introduced A to me). This in turn means that examples (14a–b) do not provide evidence against IO(DAT) > DO(ACC) because they are not actually double-object constructions. (14b) is ungrammatical simply because * … *dass man den Gästen vorgestellt hat* ' … that one introduced to the guests', where *vorstellen* has a DAT argument but is missing its obligatory ACC argument, is ungrammatical as well.

Turning to Vogel's examples (15a–b), however, which express the same meaning as (14a–b) but avoid use of *einander*, we are indeed faced with a use of *vorstellen* that works like that of *zeigen* in (1a–b)/our (2a–b), suggesting that DO(ACC) > IO(DAT) is the only grammatical order when one of the co-referent objects is the reflexive *sich*. Crucially, no mirror image scenario is involved here, so an appeal to the hair-salon-induced role reversal is not an option. We can still, however, fall back on our hypothesis in (8) and appeal to interference of inherent reflexivity. As corroborated by examples like (16), even when the reflexive is not used in its high position (here after *man* 'one') but following the other object, DAT-marking on that other object invokes subject-orientation of the reflexive.

(16) dass man      den      Gästen      nicht nur  sich  sondern auch  sein       Konzept
     that one.NOM  the.DAT  guests      not   only REFL but      also  one's.ACC concept
     vorstellen    musste.
     introduce     must
     'that one needed to introduce to the guests not only oneself but also one's concept.'

In (16), where *sich* is coordinated with a non-reflexive DP (*sein Konzept*), the inherently reflexive use of the verb, *sich jemandem vorstellen* 'introduce oneself/say one's name', is combined with the non-idiomatic ditransitive use of the verb *jemandem etwas vorstellen* 'introduce something to somebody' Given our hypothesis, in order to disambiguate between meanings (i) and (ii), i.e., to avoid meaning (ii) and thus subject-orientation of the reflexive in (15), the non-reflexive object needs to be ACC-marked.[7]

Another verb with both ditransitive and inherently reflexive uses that can be found in the literature on object coreference is ***überlassen***. Use (i) of this verb, *jemandem*.DAT *jemanden/etwas*.ACC *überlassen*, comes with the meaning 'leave someone/something (as a task) to somebody', as in (17).

(17) dass niemand      einem    Fremden   eine      wichtige  Aufgabe  überlassen  würde.
     that nobody.NOM  a.DAT    stranger  an.ACC   important  task      leave        would
     'that nobody would leave an important task to a stranger.'

The order of DO > IO sounds equally good here, but, again, this alternative word order option can easily be derived via non-case-related scrambling.

Use (ii), *sich jemandem*.DAT *überlassen* 'surrender or abandon oneself to somebody', is the inherently reflexive, idiomatic version of this verb and is illustrated in (18), where *sich* can once again occur in pre-subject position.

(18) dass sich  der       Gläubige  voll    dem       Herrn  überlässt.
     that REFL  the.NOM  believer  fully   the.DAT  Lord   surrenders
     'that a believer fully surrenders to the Lord.'

Use (iii), *jemanden*.ACC *sich*.DAT *selbst überlassen* 'leave an animate entity to its own devices' is interesting in that it also comes with idiomatic meaning but is ditransitive instead of inherently reflexive. Here, the reflexive is not subject- but object-oriented. An example is provided in (19). Both the object-orientation of *sich* and the intensification of *sich* with *selbst* 'self' make it impossible for the reflexive to occur in pre-subject position in sentences like this.

(19) dass der       Vater   die      Kinderᵢ   einfach  sichᵢ      selbst  überlässt.
     that the.NOM  father  the.ACC  childrenᵢ  simply   REFL.DAT  self    leaves
     'that the father simply left the children to their own devices.'

The ditransitive idiomatic use of *überlassen*, which requires the order of DO(ACC) > IO(DAT), may make this otherwise marked word order particularly common with this verb. In fact, Featherston and Sternefeld [16], referencing [23], give the example in (20). They note that it is better when *sich* occurs with *selbst*, but that it is not ungrammatical as is.

(20)  Hans          überlässt     die          Schwester_i   sich_i.
      Hans.NOM      leaves        the.ACC      sister        REFL.DAT
      'Hans leaves his sister to herself.'
      [16] (p. 28)

The (marginal) acceptability of this example seems to be due to a combination of the normal ditransitive use (i) and the idiomatic ditransitive use (iii) of *überlassen*. If the first object were DAT instead of ACC-marked, the inherently reflexive use (ii) would be invoked, as it is in (21).

(21)  dass man_i       dem       lieben Gott   nicht    nur      sich_i   sondern auch  seine
      that one.NOM     the.DAT   dear God      not      only     REFL     but     also  one's.ACC
      Familie          überlassen          sollte.
      Family           surrender           should
      'that people should surrender to their dear God, not only themselves but also their families.'

Thus, again, DO > IO order and therefore ACC-marking of the first object in examples like (20) might be a way to ensure expression of meaning (iii), associated with the ditransitive use, and thus avoidance of meaning (ii), associated with the inherently reflexive use.

Another verb that allows for both IO > DO and DO > IO order, similar in meaning to *überlassen*, is ***anvertrauen***. Its use (i), *jemandem*.DAT *jemanden/etwas*.ACC *anvertrauen* is ditransitive and comes with the meaning 'entrust somebody with someone/something', as in (22).

(22)  dass     niemand      einem      Fremden    ein      Geheimnis anvertrauen    sollte.
      that     nobody.NOM   a.DAT      stranger   a.ACC    secret      entrust         should
      'that nobody should entrust a stranger with a secret.'

Once again, there is a use (ii) of this verb, *sich jemandem*.DAT *anvertrauen* 'confide in somebody', that is inherently reflexive and idiomatic. As expected and shown in (23), the inherently reflexive use of the verb allows *sich* to occur in pre-subject position.

(23)  dass     sich    Teenager         selten    ihren      Eltern       anvertrauen.
      that     REFL    teenagers.NOM    rarely    their.DAT  parent       confide
      'that teenagers rarely confide in their parents.'

As with *überlassen*, if the ditransitive use (i) is intended and the second object is a reflexive, ACC-marking of the first object, as shown in (24), is the best way to push object-coreference and thereby ensure that the inherently reflexive use (ii) does not get in the way of the intended meaning.

(24)  Man          sollte    Kinder_i         nicht        sich_i       selbst       anvertrauen.
      one.NOM      should    children.ACC     not          REFL.DAT     self         entrust
      'One should not entrust children with themselves.'

If the first object is DAT-marked, the inherent reflexive use (ii) of the verb and thus subject-orientation are unavoidable, even if the reflexive occurs in a normal internal argument position, as in (25). This example, like (16) and (21), combines the subject-oriented ditransitive meaning (i) with the inherently reflexive meaning (ii).

(25)  dass die          junge        Frau_i       dem      Therapeuten  sich_i und  ihre       gesamte
      that the.NOM      young        woman        the.DAT  therapist    REFL   and  her.ACC    whole

      Lebensgeschichte   anvertraut         hat.
      life-story                entrusted          has

      'that the young woman confided in the therapist with herself and her whole life story.'

The last verb to be discussed here is ***aussetzen***, which is known for obligatory ACC>DAT order of the internal arguments in its ditransitive use (i), *jemanden/etwas*.ACC *einer Sub-*

*stanz/einem Zustand*.DAT *aussetzen* 'expose someone/something to a substance/state, as in (26).

(26) dass man niemanden der Kälte aussetzen sollte.
     that one.NOM nobody.ACC the.DAT cold expose should
     'that one should not expose anybody to the cold.'

This might appear to be evidence against IO(DAT) > DO(ACC) as the base order of all double-object constructions, but, as laid out in [19], it is not. Looking back at Figure 1, repeated here as Figure 2, with an added argument slot, it is easy to see how exceptionally patterning verbs like *aussetzen* and *unterziehen* 'cause to undergo' can be analyzed while maintaining IO(DAT) > DO(ACC) base order.

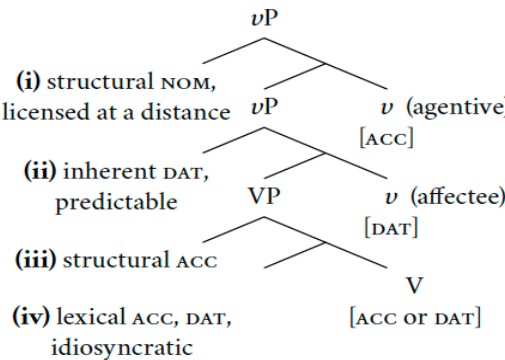

**Figure 2.** Base structure of the verbal domain including a lexical case position.

In (26), the first (ACC-marked) object is the structurally case-licensed DO in Spec VP (see position (iii) in Figure 2). There is no IO because the second object is not an inherently case-marked DAT-argument, i.e., it is not an Affectee (animate Goal, Recipient, or Source), so position (ii) in Figure 2 is not used—there is no affectee *v*P layer. The second (DAT-marked) object is a lexically (idiosyncratically) case-marked nominal in sister-to-V position (see position (iv) in Figure 2). Object-coreference with *aussetzen* or *unterziehen* is virtually impossible to construe.

Finally, use (ii) of *aussetzen*, *ein*.ACC *Lebewesen aussetzen* 'abandon/leave someone/an animal (on the street)' is monotransitive and is thus incompatible with object coreference.

To conclude this section, all the potentially ditransitive verbs discussed here, which have also been used in the literature to argue for DO(ACC) > IO(DAT) as underlying order based on object coreference binding facts like those discovered by Grewendorf, have an inherently reflexive use. This supports our hypothesis in (8), namely that the order of DO(ACC) > IO(DAT) is only acceptable because the preferred order of IO(DAT) > DO(ACC) in double-object constructions resembles the inherently reflexive use of the verb, with a DAT-marked non-reflexive object, and when this meaning is not intended, the best alternative seems to be DAT-ACC case switching, so that the non-reflexive object is ACC-marked. Unless the scenario described by the verb involves a hair salon role reversal, only the cases of the two objects are switched, not also their thematic roles. Section 5 works through an attempt at a formal account of this.

## 5. Towards a Formal Account of Case-Switching in Object Coreference Constructions

According to Bruening's theory of idiom formation [18], the subject-oriented anaphors in the examples discussed throughout this paper are generated within the VP, [VP sich_ACC verb]. This forces the interpretation of the verb as an inherently reflexive, idiomatic predicate. Thus, the structure [VP sich_ACC zeigen], when morphologically realized as marked, forces the interpretation of *zeigen* as 'appear/show oneself in public' rather than the ditransitive 'show'. We claim that the observed realizations of object coreference, which come with the interpretation of *zeigen* as 'show', exhibiting ACC > DAT order or the

addition of *selbst*, are repair strategies which are used to prevent the inherently reflexive, subject-oriented interpretation of *sich*.

In the formal implementation of this, we must take interpretation and lexical meaning to result from the output of not only LF but also PF. It is the contents of (the extended) VP which are responsible for encoding meaning differences. Based on the case-marking of the anaphor and its antecedent or the inclusion of *selbst*, the verb's lexical meaning shifts. If interpretation is read off the structure and form of the expression, then this is perhaps as expected. The Encyclopedia in Distributed Morphology [24] is a list of special/idiomatic meanings that can be associated with single lexical items (terminal nodes) or with larger structures [25,26]. This list is consulted after the output of PF and LF functions. Given the regularity of the lexical semantic force for inherently reflexive verbs, relegating their meaning to the Encyclopedia might seem concerning. However, all words and phrases in Distributed Morphology may involve Encyclopedic knowledge [25,26]. As roots themselves lack specific lexical semantic meaning, it is only in their morpho-syntactic context that they are evaluated.

In the case of German reflexive ditransitive constructions, inherently reflexive predicates have unique meaning based on the combination of V and the ACC-marked anaphor.[8] When not used ditransitively, verbs with [$_{VP}$ sich$_{ACC}$ V]-structure introduce both necessary components for interpretation within the same domain, VP. This does not prevent further movement of the anaphor, as is evident from the high position of *sich* in many of the examples in Section 4. We assume that, despite movement, the anaphor can still be interpreted locally to the verb. This may be via reconstruction based on any structure or features remaining after linearization. Encyclopedic interpretation, then, can be based on both syntactic structure and surface morphology.

If the Encyclopedia interprets [$_{VP}$ sich$_{ACC}$ zeigen] or [antecedent$_{DAT}$ [$_{VP}$ sich zeigen]], it yields the inherently reflexive, idiomatic meaning. If the intended conceptual force of the sentence is non-inherently reflexive, a crash results and the sentence will not be interpreted.

Consider the structure we propose for object co-reference in double-object constructions (DOCs) like 'show the patient herself' in Figure 3.

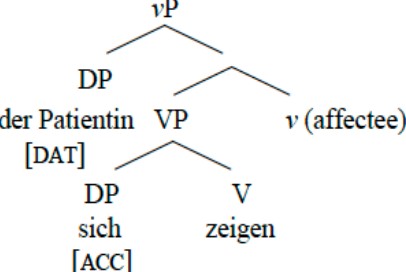

**Figure 3.** Structure for German reflexive DOC.

If this structure is reconstructed and fed into the Encyclopedia, it yields an interpretation consistent with the inherently reflexive meaning because the anaphor is interpreted to have ACC case. In order to prevent a mismatch between idiomatically assigned meaning and the conceptual force of the sentence, another form of the sentence must be selected for interpretation.

What other structures are available for interpretation? If we broaden the scope of evaluation for these options to include Encyclopedic interpretations, we can derive the variety of options available for binding in German DOCs. Within the narrow syntax, German object coreference is built as in Figure 3, where the anaphor is c-commanded by the R-expressions with which it can be intended to be coreferential. In this configuration, we are able to uphold the standard assumptions about binding theory [5]. While the exact series of operations which licenses (or builds) the anaphor may not be the same as on this early approach (see below for some discussion), we are nonetheless able to maintain the same configuration of binding. Building on other c-command diagnostics for argument

structure, like scope discussed in Section 3, we can presume this is the stable structure for introducing arguments in German.

If the structure in Figure 3 is correct and the mapping of arguments and case features is always the most direct, then we predict that [VP sich_ACC zeigen] and its inherently reflexive interpretation is the sole realization of such a structure. However, there are a variety of different realizations of reflexive DOCs. More specifically, there are three: (a) a sentence strictly faithful to the narrow syntactic representation, (b) a sentence with an element interrupting the idiomatic VP, and (c) a sentence marked morphologically to prevent reconstruction of the idiomatic VP.

Option (a) is the inherently reflexive (idiomatic) realization: [VP sich_ACC zeigen] is generated and sent to the Encyclopedia for interpretation. Given that the inherent reflexive meaning is the one intended by the speaker, the sentence is interpreted and produced. If the inherent reflexive meaning is not meant by the speaker, the interpretation of the structure will clash with the intended force, producing a crash.

Option (b) is to interrupt the idiomatic VP: [VP sich_ACC selbst zeigen] is derived by the insertion of the intensifier *selbst* after the anaphoric element.

The use of *selbst* can be due to selection from the numeration and inclusion in the derivation proper, but because *selbst* is always an optional addition to the anaphor, when non-inherently reflexive, *selbst*-insertion may be a last resort operation to disambiguate the orientation of the anaphor. This may be tied to late insertion of adjuncts [27]. Featherston and Sternefeld's quantitative generalization (c) from Section 3 states that coreference is most readily acceptable if the anaphor is intensified with *selbst* [16]. This may be the most minimal alteration to the base structure that prevents a crash at the Encyclopedia. This candidate does not produce inherently reflexive meaning because the Encyclopedic interpretation must be local. As noted in Section 4, inherently reflexive anaphors cannot be intensified by *selbst* (see also [28]). The *selbst*-structure, shown in Figure 4, interrupts the idiomatic domain in Figure 3, thus allowing for the non-inherently reflexive, ditransitive interpretation.

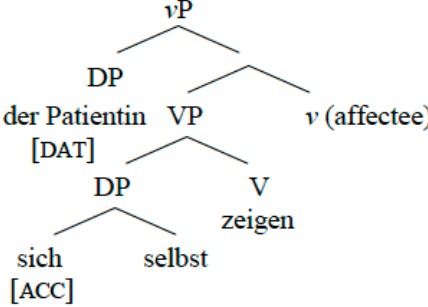

**Figure 4.** Insertion of *selbst* to interrupt the inherently reflexive (idiomatic) VP.

Note that, since the non-idiomatic/non-inherently reflexive ditransitive reading with subject-orientation of *sich* is still available in Figure 4, at least marginally, *selbst*-insertion facilitates, but does not strictly force, the desired object-coreference reading.

Option (c) involves case switching to yield [antecedent_ACC [VP sich_DAT zeigen]]. Unlike *selbst*-insertion, this repair strategy completely prevents subject orientation of *sich*, guaranteeing object coreference. Consider again (2a), our version of Grewendorf's (1a), reproduced here as (27).

(27)　dass　der　　　Arzt_i　<u>die　　　Patient**in**_j</u>　<u>sich_*i/j/ihr*_j</u>　　　im　　　Spiegel　zeigte.
　　　　that　the.NOM doctor_i　<u>the.ACC patient.**F**_j</u>　<u>REFL*_i/j/her*_j.DAT</u>　in.the　mirror　　showed
　　　　'that the male doctor showed the female patient herself in the mirror.'

The indirect object (IO)-antecedent is spelled out with morphological ACC, despite the IO's inherent DAT case assigned in the narrow syntax.

The exact mechanism for the transfer of case features is unclear, but an obvious solution to pursue is the adoption of a derivational approach to binding, that is, one on which anaphors are licensed by movement or agreement. Such an approach would allow the sharing of features along the movement or agreement chain. Movement-based approaches to anaphora (e.g., [9,10]), license or rather produce anaphors by moving a DP to a position within the same domain that c-commands its original position. Subsequently, a Spell-Out rule must be stipulated that alters the realization of a bound DP from its full R-expression to an anaphor. *John$_i$ likes John$_i$* would become *John likes ~~John~~ himself* based on a rule associated with chain reduction [29]. Agreement approaches to anaphora (see, e.g., [11]) require a phi-agreement process to license an uninterpretable anaphor. The anaphor and the antecedent enter an Agree relationship by which the phi-features are shared among the two DPs [30]. Combining both movement and agreement, Rooryck and Vanden Wyngaerd argue that the anaphor is either generated c-commanding its antecedent inside the same nominal phrase or moved to the edge of the *v*P-phase to c-command it. In need of phi-feature valuation, the anaphor probes its antecedent and enters into an Agree relation with it [12]. In all of these approaches, a syntactic relationship is formed between anaphor and antecedent. We hypothesize that it is through this relationship that the case features of anaphor and antecedent might be switched. Particularly, movement-based binding accounts might provide the Spell-Out mechanisms that allow switching of the morphological features of the two DPs or, more specifically, alternating which DP undergoes reduction to an anaphor. Thus, *zeigte der Patientin die Patientin* would be spelled out as *zeigte ~~der Patientin~~ die Patientin sich*. This option is clearly the least minimal way of spelling out a structure, certainly compared to *selbst*-insertion. Hence, it is not surprising how difficult it is to find corroborating data and thus to replicate Grewendorf's findings.

A problem with our tentatively proposed derivational account of case switching is finding a trigger for an operation which functions rather unpredictably. DAT-ACC case switching must be restricted to the very specific double-object coreference scenario being investigated here. As already explained, perhaps in an effort to prevent a crash at the Encyclopedic interface, the case switching operation applies to disambiguate the structure from its inherent reflexive meaning. The faithfully generated morphological string is produced and tested for its meaning, and, if it matches the speakers' intention, is produced. If not, a substitution of the case features applies and meaning is again tested against the speaker's intention. This substitution and testing against levels of meaning may be formulated as a phrasal application of Safir's morphological competition for anaphora [31].

Besides the rather heavy burden this analysis places on the Syntax-Encyclopedia interface, an even bigger worry is that the structure in Figure 3, when fleshed out in a derivational approach to binding [9–12] is not the right base configuration for subject orientation of the anaphor. In order for subject orientation of the anaphor to interfere with an intended object-coreference scenario, the speaker would need the flexibility of having the anaphor refer to either the IO or the subject. Given the German reflexive pronoun *sich*, which is underspecified for case, number, and gender and known for its ability to find a binder locally or across a phase boundary [32,33], this flexibility seems to be built in. However, on a movement-based derivational approach to binding, antecedent and anaphor start out as one and the same DP or at least enter into an Agree relation inside the same phrase, i.e., are necessarily coreferential. On Hornstein's approach [9], movement of the DP that antecedent and anaphor originate as creates an in situ copy and a moved copy. The latter ends up as the antecedent, and the former as the reflexive pronoun. This will be explained in more detail with the help of Figure 5 below. In Figure 3, if the speaker intends object coreference, the anaphor would have to be the result of the in situ copy of 'the patient' being replaced by the reflexive pronoun *sich*. However, it would have to be the result of replacing an in situ copy of 'the doctor' if the speaker's intention is the more common subject-object coreference. There should never be interference of the latter coreference possibility with the former because, depending on the intended meaning, either only the object DP or only the subject DP will be split into antecedent and anaphor via movement.

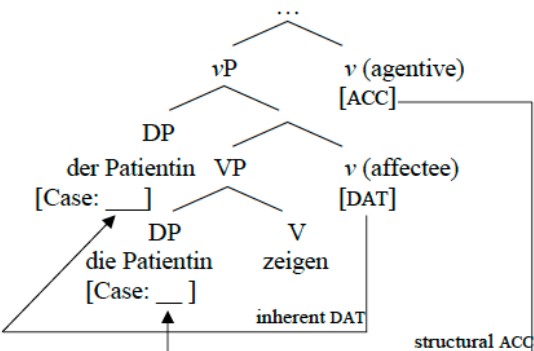

**Figure 5.** Case-licensing of the two DP-copies (lower in situ and higher remerged).

Figure 5 shows the object coreference scenario with unswitched DAT-ACC case marking of IO and DO, given a derivational analysis à la Hornstein [9]. Assuming that movement produces binding configurations, Hornstein argues for English that a DP like 'the patient' merges with *self* before moving to a higher structural position. The case of each element, the moved DP and *self*, is determined by which element receives case first. In English, *self* is assigned ACC in situ, allowing the DP to move and receive case higher in the structure. As discussed for German, anaphoric DPs do not always merge with *selbst* 'self'—*selbst* is an optional intensifier [34]. Therefore, both the moved DP and the copy left in situ have case features that need to be valued. Note that this means we have to assume that the two DP copies do not behave like the same object when it comes to being case-licensed. In Figure 5, before case-licensing happens, the DP 'the patient' is merged with the verb, and then a copy of it moves and is remerged higher, combining via second merge with affectee (applicative) *v*.

In order to prevent a moved DP from spelling out twice and creating a *\*John likes John (self)* configuration, Hornstein assumes, following Nunes [35], that lower copies are deleted at PF in order to satisfy the conditions of Kayne's Linear Correspondence Axiom [36]. In English, this strands the *self*-morpheme, which triggers the insertion of a pronoun to host the bound morpheme. In German, we might assume that NP/DP levels are deleted but strand certain features, like the case feature. This then requires insertion of the reflexive pronoun *sich*.

To get the desired ACC-DAT order of case-marked internal arguments, we can appeal to the possibility of the higher DAT-marked copy of the two DPs being deleted and replaced by *sich*, producing an instance of backward control. This would need to be followed by movement of the lower ACC-marked DP above the DAT-marked *sich*. This instance of backward control coupled with obligatory movement of the lower object above the higher object, as shown in Figure 6, has to be motivated, of course.

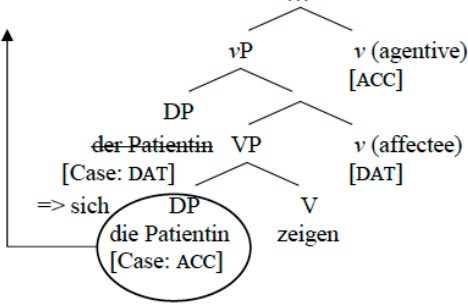

**Figure 6.** Deletion and replacement of higher copy followed by movement of lower copy.

Given the analysis argued for in Section 4, the motivation for these operations should be interference of the inherently reflexive interpretation of the verb and thus subject orientation of the anaphor. But this brings us back to the problem of trying to fit our German

object coreference facts into a derivational account of binding. Subject orientation of the anaphor cannot interfere with object coreference if it is a DP ending up in an object position from which the anaphor is derived and with which it is therefore automatically co-referent.

## 6. Conclusions

As already pointed out by Featherson and Sternefeld, the difficulty of analyzing Grewendorf's data [4] stems from idiosyncratic variation in the generation of German object coreference constructions [16]. The investigation embarked on in this paper has led us to a single base structure of IO > DO, corresponding to the canonical surface order, in the narrow syntax. At the same time, our investigation points toward the need for a variety of Spell-Out options, including one where DAT/ACC-case-marking is switched. The attempted formal account sketched to capture this, a movement-based binding analysis à la Hornstein [9], runs into a previously unnoticed problem for derivational approaches to binding more generally. It appears that the German anaphor *sich* needs to be underspecified and thus reference-free (at least) at the beginning of the derivation, meaning it cannot start its life as a part of, or standing in for, a specific DP, and it also cannot immediately enter into a local Agree relation with a DP to get its phi-features valued. Thus, for German binding between objects (and possibly other binding phenomena), Chomsky's classic Binding Theory [5,6], consisting of at least Conditions A and B, applying as a filter upon the completion of each phase [32,33] to check that each anaphor and pronominal is properly bound or free, is a better solution than a derivational approach to binding. Of course, this leaves the non-canonical ACC>DAT order unaccounted for, and we agree with the authors of all the derivational binding accounts discussed here [10–13] that binding can only be made fully compatible with the Minimalist Program [37] if there are no binding-specific elements like indices or features representing reflexivity as a primitive of the theory. Clearly, some tweaking of derivational binding approaches needs to be done before German object coreference can be fully captured within Minimalism.

**Author Contributions:** Conceptualization, V.L.-S. and N.T.; formal analysis, N.T. and V.L.-S.; writing—original draft preparation, V.L.-S.; writing—review and editing, V.L.-S.; project administration, V.L.-S. All authors have read and agreed to the published version of the manuscript.

**Funding:** This research received no external funding.

**Institutional Review Board Statement:** Not applicable.

**Informed Consent Statement:** Not applicable.

**Data Availability Statement:** Not applicable.

**Conflicts of Interest:** The authors declare no conflict of interest.

## Notes

[1]   Although the verbs we label "inherently reflexive" do allow for the reflexive pronoun to be replaced with a referential DP and could thus be labeled "naturally reflexive" instead [8], we stick with "inherently reflexive" because the idiomatic meaning that comes with the reflexive use of these verbs, which is crucial for our analysis, is unavailable when the reflexive pronoun is replaced with a referential DP. Thus, in Section 4, we consistently refer to the verbs' use as inherently reflexive when idiomatic meaning is involved, and as ditransitive when they involve a non-idiomatic double-object construction where one of the objects happens to be a reflexive pronoun.

[2]   We judge the pronominal options as degraded (the first author of this paper is a native speaker of German), but Grewendorf marks the non-reflexive pronoun in (1b) as grammatical.

[3]   As noted by an anonymous reviewer, our interpretation of the generalizations in (6) may be over-reaching. The fact that the intensifier *selbst* is preferred in our sentences in (3), for example, even when the order of arguments is the canonical IO(DAT) > DO(ACC) one, suggests that these sentences are marginal, not fully grammatical, anyway. It is thus not clear that the relationship

between degradation of acceptability and changing of canonical structure is a causal one. This point is well-taken, but the fact that our sentences in (3a–c) are clearly better in the canonical IO(DAT) > DO(ACC) order than in the non-canonical DO(ACC) > IO(DAT) order still allows us to establish a connection between [16]'s findings and our data in a way that supports our line of argumentation.

4    By "Affectee *v*P", we mean what is often labeled as "ApplP/*v*applP" introducing an applicative argument (Recipient, Beneficiary, etc.).

5    This judgment, (10b) being acceptable in a hair salon scenario, where the speaker *I* is the hair stylist, comes from the second author of this paper, who is a native speaker of English, and it was confirmed by a native-English-speaking hair stylist. Note, however, that an anonymous reviewer, also a native speaker of English, disagrees with this judgment, backing up Vogel's.

6    The translations of the various verb uses and the grammaticality judgments on the given examples in this section are based on the native speaker intuitions of the first author.

7    This argumentation is more immediately convincing when the subject is 3rd person SG/PL (not 1st person as in (15)) because of agreement with *sich*, but a verb that has an inherent reflexive use and occurs with a non-reflexive DAT-argument may signal subject-orientation of the reflexive, even when there is a person mismatch between subject and anaphor.

8    Although it is clear that the third person anaphor *sich* does not display any morphological distinctions between its DAT and ACC forms, we still assume the ACC-marking of the anaphor is interpreted in the context of a DAT-marked antecedent. *zeigen* is not a double-ACC verb (see [19]) and is therefore expected to have differently case-marked internal arguments.

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
