# Peer review of "Object Coreference in German: The Reflexive sich as a Problem for Derivational Approaches to Binding"

_philosophies, doi:10.3390/philosophies7010005_

Round 1
Reviewer 1 Report
The paper argues in favour of there being just one base order for the double object construction in German, with the Indirect Object (IO) c-commanding the Direct Object (DO). The authors present evidence against the reverse order of arguments proposed by Grewendorf (1988) and Vogel (2014). They argue that the order DO>IO results from the avoidance of idiomatic inherently reflexive interpretation of the verb involved. The analysis is well-, in presented, all the arguments are sound.
I want to mention two main points that to me require some elaboration in the paper. The first one concerns case marking, in particular the question how structural case is assigned/valued. On p. 5, we learn that v licenses structural ACC, but how is it done - via Agree or as a dependent case? Later on, in Section 5, case switching in DO>IO has been appealed to. Once again, the question arises how this is done. The authors suggest that the V may assign either lexical accusative or lexical dative. I have doubts concerning this claim, as lexical case is idiosyncratic, and one verb normally assigns one lexical case.
Another problematic issue concerns the status of the reflexive marker itself. On p. 5, it is written that it is not an argument because it cannot be intensified by 'selbst'. For me, this is no enough, because the lack of intensification might simply indicate that 'sich' is a clitic, not necessarily a non-argument. On p. 9, in turn, 'sich' is intensified although the verb has an inherently reflexive interpretation, mentioned by the authors. In footnote 6, on page 12, 'sich' with inherently reflexive verbs is treated as an argument with lexical accusative case. So, one is left to wonder what exactly is the status of the reflexive marker 'sich' in the structures analysed in the paper.
The last minor point concerns the remark made on p. 7 that 'einander' is just an adjunct, not an argument. Is there any evidence to support this claim. The authors just refer to Sternefeld and Featherson's (2003) work, but I think they could provide some evidence here.
Two works mentioned in the text do not appear in the References, i.e. (i) McFadden (2006) p. 4, and (ii) Jackendoff (1992) p. 8
Author Response
All major changes I made in response to our reviewers' concerns are highlighted in yellow in the revised document. Here, I explain how I addressed Reviewer 1's comments.
Regarding the question how structural case is assigned/valued, I changed the wording on p. 5 under Figure 1 a bit to make clear that it is functional heads licensing case / valuing case features on DPs. We're not dealing with dependent case here.
As for our case switching proposal, in section 5, on p. 14 of the revised paper, we make clear that the exact mechanism for the transfer of case features is unclear, but that we take a derivational approach to binding to sketch a possible account. On this account, we hypothesize that DAT and ACC can be switched on the two internal argument DPs via feature sharing. Ultimately, we conclude, of course, that a derivational approach to binding is incompatible with the German object coreference facts discussed in the paper, so the case switching idea can't be pursued further.
Concerning Figure 2 right above section 5 on p. 11 of the revised paper, the reviewer seemed to have misunderstood the lexical ACC or DAT case assignment claim. We don't say that a given verb can assign either case. We say that, depending on the V, it may come with lexical ACC or lexical DAT case. This is based on Lee-Schoenfeld 2018 and explained in detail there. Lexical case doesn't play a role in the case switching idea.
The point regarding the nature of the reflexive marker sich is well-taken but not the focus of this paper. Because our data and conclusions don't directly shed light on the argument status of the different uses of sich, I omitted the sentence about sich not being an argument in its high position in (7), (2'b) and (2''b). This sentence was on p. 5 in the draft the reviewer commented on but would have been on p. 6 of the revised paper, right under (2''b).
As for einander being an adjunct rather than an argument, as claimed by Sternefeld & Featherston (2003), I added a passage on p. 8 of the revised paper to provide further evidence for their claim.
McFadden 2006 and Jackendoff 1992 have been added to the references.
Thank you!
Reviewer 2 Report
General commentary
This is an interesting response to the problem posed in Grewendorf (1988), and the hypothesis posed in (8) gives the authors a great amount of traction. Generally the data are convincing. However, three major comments and some desiderata are presented below.
- The interpretation of Featherston & Sternefeld (2003) is over-reaching. Higher acceptance of any structure could be due to a myriad of reasons, and the fact that the intensifier selbst is preferred suggests that most of the structures in sections 1-3 are marginal anyway. The authors, in fact, are hinting that the IO>DO versions of (3a-c) are not fully grammatical (indeed, they are only “better” (line 123) than then DO>IO versions). If canonical options are intended to be the most highly accepted, then what has been “changed” from the ideal form to the still degraded IO>DO versions of (3a-c)? Note that this critique is not related to whether degradation of acceptability exists so much as the implication that the relationship between degradation and changing canonical structure is so clearly a causal one.
- As a native speaker of English, I agree with Vogel’s (2014) judgment of (10b) even in a hair salon (i.e., that (10b) is impossible). Given that the authors dismiss the hair salon case as the exception at the beginning of section 4.1, the reader wonders whether it needs to be included.
- In section 5, some discussion of an alternative to the derivational account (or a modification of its assumptions) is needed.
Desiderata:
-
Lines 71-72 – What about V2 would be problematic in these examples?
-
Lines 176-178 – Why is the agent accusative in Figure 1?
-
Lines 534-546 – Figure 3 illegible.
-
Lines 590-600 – Figure 4 illegible.
-
Page 16 – Figures 6 and 6 illegible.
-
Jackendoff (1992) missing in References.
Author Response
All major changes I made in response to our reviewers' concerns are highlighted in yellow in the revised document. Here, I explain how I addressed Reviewer 2's comments.
- I changed the wording of the passage under (6) on p. 4 of the revised paper to make clear that Featherston & Sternefeld's (2003) generalizations are just tendencies that may exist for a number of reasons not taken into account in our paper. However, I kept our interpretation of F & S's generalizations and added a footnote (fn 3) to address the reviewer's concerns about assuming that the relationship between degradation of acceptability and changing of canonical structure is a causal one.
- Here, I added a footnote as well (fn 5 on p. 6 of the revised paper), explaining our grammaticality judgment and acknowledging that other native speakers of English, like the reviewer, may side with Vogel and thus disagree with our judgment. I do consider the strange mirror image role reversal important in the context of putting Grewendorf's famous 'doctor showing patient in the mirror' example into perspective, so I did not want to cut this discussion from the paper.
- I added some discussion along these lines to the conclusion (see p. 16 of the revised paper). How derivational approaches to binding can be tweaked to allow for an account of German object coreference isn't clear yet, but I emphasized that this is the direction that future research has to take if we want to make our analyses of binding phenomena compatible with the Minimalist Program.
- Desiderata: I changed the wording that explains the rearrangement of Grewendorf's examples from main clause into embedded clause order (see lines 70-71 of the revised paper). In Figure 1, the agent is not accusative. I tried to clear up this misunderstanding (see lines 178-181 of the revised paper). Figures 3, 4, 5, and 6 must have been illegible because they were submitted in Arbor Win font. I've fixed them by pasting in snapshots of them from the PDF version of the paper. The Jackendoff 1992 reference has been added.
Thank you!